

# Chemical characterisation of potential pheromones from the shoulder gland of the Northern yellow-shouldered-bat, *Sturnira parvidens* (Phyllostomidae: Stenodermatinae)

Chris G. Faulkes[1], J. Stephen Elmore[2], David A. Baines[3], Brock Fenton[4], Nancy B. Simmons[5] and Elizabeth L. Clare[1]

[1] School of Biological & Chemical Sciences, Queen Mary University of London, London, UK
[2] Department of Food and Nutritional Sciences, University of Reading, Reading, UK
[3] Baines Food Consultancy Ltd, Thornbury, Bristol, UK
[4] Department of Biology, University of Western Ontario, London, ON, Canada
[5] Department of Mammalogy Division of Vertebrate Zoology, American Museum of Natural History, New York, NY, USA

Corresponding author
Chris G. Faulkes,
c.g.faulkes@qmul.ac.uk

## ABSTRACT

Bats of the genus *Sturnira* (Family Phyllostomidae) are characterised by shoulder glands that are more developed in reproductively mature adult males. The glands produce a waxy secretion that accumulates on the fur around the gland, dyeing the fur a dark colour and giving off a pungent odour. These shoulder glands are thought to play a role in their reproductive behaviour. Using gas chromatography–mass spectrometry, we analysed solvent extracts of fur surrounding the shoulder gland in the northern-shouldered bat, *Sturnira parvidens* to (i) characterise the chemical composition of shoulder gland secretions for the first time, and (ii) look for differences in chemical composition among and between adult males, sub-adult/juvenile males and adult females. Fur solvent extracts were analysed as liquids and also further extracted using headspace solid-phase microextraction to identify volatile components in the odour itself. Odour fingerprint analysis using non-metric multidimensional scaling plots and multivariate analysis revealed clear and significant differences ($P < 0.001$) between adult males vs both juvenile males and adult females. The chemical components of the shoulder gland secretion included terpenes and phenolics, together with alcohols and esters, most likely derived from the frugivorous diet of the bat. Many of the compounds identified were found exclusively or in elevated quantities among adult (reproductive) males compared with adult females and non-reproductive (juvenile) males. This strongly suggests a specific role in male–female attraction although a function in male–male competition and/or species recognition is also possible.

## INTRODUCTION

Pheromones are intra-specific semiochemicals (chemical signals) that modify the behaviour and/or physiology in the recipient animal. They occur across the animal kingdom and are particularly well characterised in insects. Among mammals, there are many examples of pheromonal communication (see *Wyatt, 2014* for recent review), but their chemical composition and actions in mammals have been debated (*Doty, 2010*). There are two general categories of semiochemicals within a species: pheromones, chemical signals which elicit a stereotypical behavioural or physiological response in the recipient animal, while signature mixtures are odour cues which are learned by the conspecific receivers and often have complex and variable chemical profiles. These 'individual mixtures' or 'signature odours', which act in social communication though learning may identify a specific individual or social group (*Wyatt, 2014*, *2017*; *Dehnhard, 2011*).

Bats (order Chiroptera), are the second most speciose mammalian order (after Rodentia), yet relatively little is known about the possible role of pheromonal communication in this group. Bats exhibit a wide variety of social and mating systems, implying the need for individual recognition within groups, and exchange of social and reproductive cues (including mate choice). A nocturnal lifestyle and potentially limited visual cues, particularly in roosts, could exaggerate the importance of auditory and olfactory cues. Chemical cues are known to be used by big brown bats *Eptesicus fuscus* (*Bloss et al., 2002*), and implicated in Bechstein's bats, *Myotis bechsteinii* (*Safi & Kirth, 2003*) and the fisherman bat, *Noctilio leporinus* (*Brooke & Decker, 1996*).

While the exquisitely complex auditory systems associated with echolocation in bats have been studied for years, work on their olfactory systems is more limited. Both the primary and secondary olfactory systems are involved in pheromonal chemoreception, with the latter forming the vomeronasal organ (VNO). A comparative anatomical study by *Wible & Bhatnagar (1996)* looked across more than a hundred genera of bats and found extensive variation in the presence or absence of the VNO, implying convergent gains and losses of the structure. More recently, genomic studies have investigated the genes coding for olfactory receptor proteins. *Young et al. (2010)* reported that two bat species (the little brown bat, *Myotis lucifugus*, and the flying fox, *Pteropus vampyrus*) completely lacked intact genes for functional receptors (V1R) in the VNO and they identified only inactive pseudogenes. Furthermore, inactivating mutations in the vomeronasal signal transduction gene Trpc2 were also characterised, suggesting that any VNO or secondary olfactory epithelia (if present) would be non-responsive to pheromonal signals. *Yohe et al. (2017)* used Trpc2 as a molecular marker for examining evolutionary losses and gains of the VNO system in more than 100 species across 17 of the 21 extant bat families. They found that all bats examined exhibit degraded vomeronasal systems, except for some members of Miniopteridae and Phyllostomidae. The *Sturnira lilium* species complex, which includes our study species *Sturnira parvidens*, does have a VNO and does not have a pseudogenisation in Trpc2 exon 2, which would prevent signal transduction (*Yohe et al., 2017*), suggesting a functional VNO and a potential role for pheromonal signalling in this group. Collectively, these anatomical and genetic studies indicate that some, but not all, bat

species have the anatomical and physiological substrates required in the accessory olfactory system for potentially receiving and responding to pheromones. However, pheromones in mammals are also detected by the main olfactory system (*Wyatt, 2014*, *2017*), so the lack of functioning VNO receptors does not rule out the possibility of semiochemical signalling.

Evidence for olfaction acting as a sensory modality for individual discrimination in bats has also been gathered from behavioural studies. Scent-choice experiments by *Bouchard (2001)* demonstrated that two African molossid bats (*Mops condylurus* and *Chaerephon pumilus*) can distinguish between the sexes and distinguish roost-mates from strangers using olfactory cues. A number of species in the family Emballonuridae have sacs in their wing or tail membranes, which may function in a scent-based identification social system. These sacs are typically larger and better developed in adult males. In the best-studied example by *Voigt & Von Helversen (1999)*, male *Saccopteryx bilineata* keep harems of females and use their paired wing sacs (located near the forward edge of each wing) to store bodily fluids. These fluids are then sprayed on females during hovering or used in characteristic 'salting' behaviour. Females also possess wing sacs but they are rudimentary and the same behaviours have not been observed (*Voigt & Von Helversen, 1999*).

Despite observations that imply important roles for olfaction in social behaviour of bats, there are few papers that identify the chemical make-up of chiropteran pheromones. *Bloss et al. (2002)* found that female big brown bats, *E. fuscus* (Family Vespertilionidae) use chemical cues to distinguish among female conspecifics, and these authors potentially identified 14 compounds based on analysis of their retention index on the gas chromatograph. Similarly, *Caspers, Franke & Voigt (2008)* and later *Schneeberger et al. (2016)* established that the odour 'fingerprints' presented to females by male *Saccopteryx bilineata* contained differences associated with age, colony, and year of sample collection. Thus, these signals could provide information to females for assessing potential male mates. However, *Schneeberger et al. (2016)* did not report the actual chemical composition of the wing-sac liquids. Other species of bats have different structures that seem to produce odoriferous secretions possibly used in social communication. Many species of molossids as well as the phyllostomid *Trachops cirrhosis* have a single gland in the ventral midline of the throat which is found either exclusively in males or is best developed in adult males (*Bowles, Heideman & Erickson, 1990*; *Scully, Fenton & Saleuddin, 2000*; *Tandler, Nagato & Phillips, 1997*; *Phillips, Tandler & Pinkstaff, 1987*). Male *N. leporhinus* (Family Noctioionidae) secrete a strongly smelling sticky substance along their lateral fur (N Simmons, 2017, personal observation). The chemical composition of these secretions has never been investigated.

The shoulder glands of *Sturnira* species are well known but understudied. *Sturnira* are widely distributed and abundant from north-western Mexico, through Central America and into tropical and subtropical South America (to the north of Argentina and Uruguay) and the Lesser Antilles (*Gannon, Willig & Knox Jones, 1989*; *Velazco & Patterson, 2013*, *2014*). The common name of this genus (yellow-shouldered bats) refers to glandular scent organs on the shoulder that produce a yellowish or reddish staining of the fur that surrounds them (Fig. 1). These glands are most highly developed in males, and

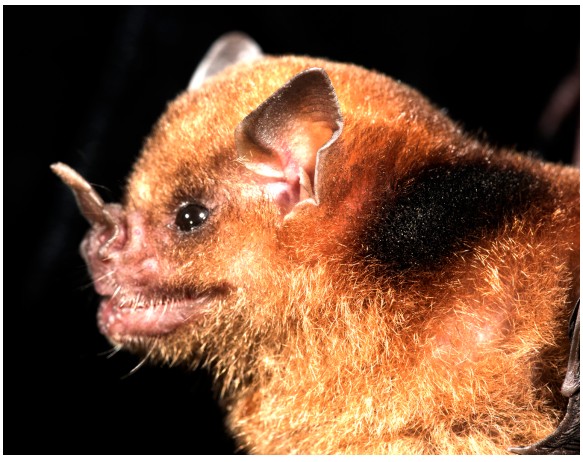

**Figure 1** *Sturnira parvidens* **adult male showing dark staining of fur surrounding the shoulder gland (Photo credit: Brock Fenton).**

development of these glands appears to be dependent on the reproductive maturity of the individual (*Scully, Fenton & Saleuddin, 2000*). In mature males they are described as emitting a 'strong, sweetish, musky odour' (*Gannon, Willig & Knox Jones, 1989*; *Goodwin & Greenhall, 1961*) that can be so strong it is detectable by humans several metres away (E Clare, 2016, personal observation).

Histological examination of the shoulder glandular region of male *Sturnira* revealed numerous hair follicles and associated sebaceous glands, with scattered small sudoriferous glands in a deeper layer of adipose tissue. The same glandular elements were also present in females, although the size of the glandular region varied between males and females. Females lack both the strong odour and strongly coloured hairs stained from secretion (*Scully, Fenton & Saleuddin, 2000*). The production of a strongly-smelling secretion in reproductively mature males implies a potential pheromonal role for these odours in reproduction, although very little has been reported about the social or mating system of these species, and no behavioural or chemical analysis has been done.

Our goal was to chemically characterise, for the first time, the secretions from the shoulder gland of the northern yellow-shouldered bat, *Sturnira parvidens*. Using a gas chromatography–mass spectrometry (GC–MS) approach, we identify the constituents and compare the odour profiles among adult males, juvenile (non-reproductive) males, and adult females.

# METHODS

## Sample collection and storage

We conducted field work and collected samples in 2015 and 2016 at sites in Orange Walk District, Belize, including the Lamanai Archaeological Reserve and adjacent secondary forest and gardens near the Lamanai Outpost Lodge (17.75117N, −88.65446W), and a forest fragment at the Ka'kabish Archaeological Project (17.8147N, 148 −88.73052W). Our research was conducted in accordance with accepted standards for humane capture and handling of bats published by the American Society of Mammalogists

(*Sikes, Gannon & the Animal Care Use Committee of The American Society of Mammalogists, 2016*) and approved Institutional Animal Care and Use Committee protocols (Brown University IACUC 1205016 and 1504000134). Locally, the fieldwork was carried out with Belize Forestry Department Scientific Research and Collecting Permits CD/60/3/15 (20) and WL/1/1/16 (26), respectively. All samples were collected during a 10-day field trip at the end of April and the start of May to control for any potential seasonal variation. We captured bats using mist nets set along forest paths, and each individual was briefly contained in a cloth 'bat bag' for identification and processing. We determined sex and then checked for age and reproductive maturity in males by examining the genitalia and ossification of the wing joints, following methods described by *Racey (1988)* and *Anthony (1988)*: males were classified as reproductive based on the size and descent of testes into the scrotal sac, and as an adult if wing joints were ossified. We removed a small sample of fur of the same size (approximately four $mm^2$) from an equivalent area surrounding the shoulder gland on all individuals, using sterile scissors. More specifically, fur was collected from a discoloured patch where oil was being secreted in adult males (Fig. 1), while for adult females and juveniles we collected fur from the same physical location as the adult males (which is often also slightly discoloured). After fur was collected, we released all bats at the site of their capture. Each fur sample was stored in a two mL screw-topped Eppendorf tube containing one mL of glyceryl trioctanoate (Sigma-Aldrich, Gillingham, Dorset, UK), a medium-chain triglyceride (MCT), at ambient temperature for later chemical analysis (within 2 months). The MCT acts as a solvent, stabilising and dissolving the organic components of any secretions from the shoulder gland.

## Direct injection of MCT extracts

Medium-chain triglyceride extracts were analysed by GC–MS, using an Agilent 7693 autoinjector and Agilent 6890N gas chromatograph with 5975 mass spectrometer (Agilent, Santa Clara, CA, USA). The GC column used was a 15 m × 0.25 mm HP-5MS UI column (0.25 μm film thickness; Agilent, Santa Clara, CA, USA) and injection was in split mode (10:1) at 250 °C, the injection volume being one μL. The initial temperature of the GC oven was 100 °C (0 min), rising at 5 °C per min to 320 °C. Helium was used as the carrier gas at a constant flow rate of 1.1 mL/min. A series of *n*-alkanes ($C_5$–$C_{30}$) in diethyl ether were analysed, under the same conditions, to obtain linear retention index (LRI) values for the components of the extract.

The mass spectrometer operated in electron impact mode with an electron energy of 70 eV, scanning from *m/z* 29 to *m/z* 800 at 1.9 scans/s. We identified compounds by first comparing their mass spectra with those contained in the NIST/EPA/NIH Mass Spectral Database or in previously published literature. Wherever possible identities of compounds were confirmed by comparison of LRI values, with either those of authentic standards or published values.

## Headspace solid-phase microextraction

We carried out volatile compound analysis by automated headspace solid-phase microextraction (SPME), followed by GC–MS, using an Agilent 110 PAL injection system

and Agilent 7890 gas chromatograph with 5975C mass spectrometer. The SPME fibre stationary phase was composed of 75 µm divinylbenzene/Carboxen™ on polydimethylsiloxane (Supelco, Bellefonte, PA, USA). The uncapped two-mL vials containing the MCT extracts were placed in 20-mL headspace vials. The samples were then equilibrated for 10 min at 37 °C before being extracted for 30 min. Samples were agitated at 500 rpm (5 s on, 2 s off) during equilibration and extraction. After extraction, the contents of the fibre were desorbed in splitless mode at 250 °C onto the front of a Stabilwax DA fused silica capillary column (30 m, 0.25 mm i.d., 0.50 µm film thickness; Restek, Bellefonte, PA, USA). The GC temperature programme and the fibre desorption step commenced at the same time. During the desorption period of 45 s, the oven was held at 40 °C. After desorption, the oven was held at 40 °C for a further 255 s before heating at 4 °C/min to 260 °C, where the temperature was maintained for 5 min. Helium was used as the carrier gas at a constant flow rate of 0.9 mL/min. A series of $n$-alkanes ($C_5$–$C_{22}$) in diethyl ether was analysed, under the same conditions, to obtain LRI values for the components of the extract. To check for the presence of volatiles contributed from the MCT solvent, we ran a blank extraction of the MCT alone within the sample vial. We detected some octanoic acid (this was predicted, as the MCT used was glyceryl trioctanoate), that we note in the results.

The mass spectrometer operated in electron impact mode with an electron energy of 70 eV, scanning from $m/z$ 20 to $m/z$ 280 at 1.9 scans/s. As with the direct injection protocol, we identified compounds by first comparing their mass spectra with those contained in the NIST/EPA/NIH Mass Spectral Database or in previously published literature. Wherever possible we then confirmed identities by comparison of LRI values with those of authentic standards.

## Chemical confirmation of compounds identified

Authentic standards for confirmation of identities and/or synthesis included 3,4-dimethoxybenzoic acid (veratric acid) (I), 4-(4-methoxyphenyl)butyric acid (II), β-(4-hydroxy-3-methoxyphenyl)propionic acid (III), 3-(3,4-dimethoxyphenyl)propionic acid (IV), linalool oxide (pyranoid), 3,4-dimethylbenzaldehyde, 1-methoxy-2-propanol, 1-propanol and acrylic acid, all purchased from Tokyo Chemical Industry UK, Ltd (Oxford, UK); 3-(3,5-dimethoxy-4-hydroxyphenyl)propionic acid (V) was purchased from Carbosynth (Compton, UK); β-(3-hydroxy-4-methoxyphenyl)propionic acid (VI), anthranilic acid (VII), vanillic acid (VIII), octanoic acid (IX), salicylic acid (X) and propylene glycol were obtained from Sigma-Aldrich (Gillingham, Dorset, UK).

To confirm the identities of tentatively identified propyl and hydroxypropyl esters, we performed synthesis using acids I to X above. A total of 10 reactions were carried out in one-mL Reacti-Vials (Thermo Fisher Scientific, Waltham, MA, USA). Each Reacti-Vial contained 0.2 g of acid and 0.2–0.6 mL of a 1:1 mixture of 1-propanol and propylene glycol plus 1 drop of hydrochloric acid as catalyst. The amount of alcohol added depended on the solubility of the acid. Reaction mixtures were heated for 30 min at 80 °C and then neutralised by dropwise addition of 0.1M sodium bicarbonate. The esters were then transferred to seven-mL glass bottles and two mL of diethyl ether were added. The contents

of the bottles were then stirred at 700 rpm using a magnetic stirrer for 5 min. The contents were allowed to settle and the ether layer was removed for analysis by GC–MS, after addition of a small amount of anhydrous sodium sulphate to remove any water still present. The GC–MS conditions used were the same as those used for the direct injection of the MCT extracts, only with a split ratio of 100:1.

## Statistical analysis

We identified GC peaks and calculated their retention index and peak areas using the software MSD Chemstation E02.02.1431 (2011; Agilent, Santa Clara, CA, USA). For direct injection and headspace analyses, we compared both mean peak areas from the GC output and the relative peak area (%) of each compound to the total peak area (of all peaks we identified) among adult males (of reproductive age), subadult/juvenile males (non-reproductive), and adult females. Absolute mean peak areas give an indication of quantitative differences between samples and groups but can be subject to variance in sampling. The percentage peak areas normalise the data across samples, and correct for differences in absolute amounts of starting sample (due to the difficulties in taking an equivalent amount of fur and secretion from each bat). In our analyses, we examined both mean and relative (%) peak areas. Taking an approach used by *Schneeberger et al. (2016)*, we compared the overall chemical compositions (the 'odour fingerprint') of each individual sample. A Bray–Curtis dissimilarity index was then computed using adult males, juvenile males and adult females as categories for prediction in a non-parametric analysis of similarities (ANOSIM) and to identify significant differences among groups. We visualised data in two dimensions with a non-metric multidimensional scaling (NMDS) ordination. Similarity percentage (SIMPER) was then used to identify significant differences between equivalent peaks in each sample, and, hence, which odour components contributed most to the differences between groups. We performed all analyses using the VEGAN package in RStudio (version 1.0.136; *R Development Core Team, 2015*), and both the ANOSIM and SIMPER procedures used 1,000 permutations.

## RESULTS

We identified 15 compounds in MCT extracts of fur surrounding the shoulder gland of *Sturnira parvidens*, some of which differed significantly between adult males and juvenile males/adult females ($n = 5$ samples from each age/sex class; Table 1; Fig. S1). Three of the 15 compounds occurred only in adult males, namely vanillic acid, 3,4-dimethoxybenzoic acid (veratric acid), and 3-(3,4-dimethoxyphenyl)propionic acid. We found two other compounds in both adult males and females (levels significantly higher in males), but not juvenile males: hydroisoferulic acid propyl ester and 3-(3,5-dimethoxy-4-hydroxyphenyl) propionic acid propyl ester. Four additional compounds showed significant differences between the three groups, and in all cases the adult males had higher levels of these compounds, while juvenile males did not differ from females. Of these four, α,β-dihydroferulic acid propyl ester had the greatest peak area; the other compounds were the propyl ester of anthranilic acid, 4-(4-methoxyphenyl)butyric acid, the propyl ester of α,β-dihydroferulic acid, and the hydroxypropyl ester of α,β-dihydroferulic acid. A single

**Table 1 Compounds in medium-chain triglyceride extracts of fur surrounding the shoulder gland of *Sturnira parvidens*.**

| Peak number | LRI | Compound | Gas chromatographic peak areas (×10⁻⁶) *relative peak areas (%)* | | | P |
|---|---|---|---|---|---|---|
| | | | Adult male | Juvenile male | Adult female | |
| 1 | 1,386 | anthranilic acid | 3.95 (1.78) | 0.161 (0.212) | 0.153 (0.262) | NS |
| | | | *4.92 (3.73)* | *10.92 (12.93)* | *5.09 (7.59)* | NS |
| 2 | 1,503 | anthranilic acid propyl ester | 14.7 (11.4)[b] | 0.214 (0.234)[a] | 0.186 (0.126)[a] | <0.05 |
| | | | *11.86 (3.28)* | *15.59 (11.21)* | *18.69 (7.98)* | NS |
| 3 | 1,552 | vanillic acid | 0.429 (0.061) | — | — | <0.001 |
| | | | *0.72 (0.70)* | — | — | <0.0001 |
| 4 | 1,626 | 3,4-dimethoxybenzoic acid (veratric acid) | 1.46 (0.945) | — | — | <0.001 |
| | | | *1.43 (0.67)* | — | — | <0.0001 |
| 5 | 1,666 | 4-(4-methoxyphenyl)butyric acid | 12.6 (16.8)[b] | 0.0107 (0.0239)[a] | 0.0335 (0.0749)[a] | <0.001 |
| | | | *6.78 (3.07)[b]* | *0.67 (1.50)[a]* | *0.94 (2.10)[a]* | <0.01 |
| 6 | 1,671 | vanillic acid propyl ester | 5.15 (6.42) | 0.0058 (0.0129) | 0.0190 (0.0426) | NS |
| | | | *2.18 (2.11)* | *0.82 (1.83)* | *0.53 (1.19)* | NS |
| 7 | 1,682 | anthranilic acid hydroxypropyl ester | 3.30 (2.54) | 0.0570 (0.109) | 0.054 (0.0741) | NS |
| | | | *2.48 (0.57)* | *2.38 (4.19)* | *2.14 (3.09)* | NS |
| 8 | 1,722 | β-(4-hydroxy-3-methoxyphenyl)propionic acid (α,β-dihydroferulic acid) | 3.73 (2.77)[a] | 0.298 (0.262)[a,b] | 0.172 (0.174)[b] | <0.05 |
| | | | *3.44 (1.51)* | *34.56 (33.65)* | *2.14 (3.09)* | NS |
| 9 | 1,741 | β-(3-hydroxy-4-methoxyphenyl)propionic acid (hydroisoferulic acid) | 0.802 (0.648) | 0.0147 (0.0223) | 0.0049 (0.0108) | NS |
| | | | *0.72 (0.41)* | *0.81 (1.37)* | *1.26 (2.81)* | NS |
| 10 | 1,744 | veratric acid propyl ester | 0.401 (0.233) | 0.0063 (0.0140) | 0.0155 (0.0346) | NS |
| | | | *0.42 (0.24)* | *0.39 (0.88)* | *0.43 (0.97)* | NS |
| 11 | 1,764 | 3-(3,4-dimethoxyphenyl)propionic acid | 0.711 (0.621) | — | — | <0.001 |
| | | | *0.54 (0.16)* | — | — | <0.0001 |
| 12 | 1,828 | α,β-dihydroferulic acid, propyl ester | 96.3 (95.7)[b] | 0.427 (0.416)[a] | 0.688 (0.712)[a] | <0.001 |
| | | | *60.21 (16.39)[a]* | *33.38 (26.14)[b]* | *49.43 (11.71)[ab]* | <0.06 |
| 13 | 1,852 | hydroisoferulic acid, propyl ester | 3.16 (3.19)[b] | — | 0.0233 (0.0412)[a] | <0.01 |
| | | | *2.03 (0.74)* | — | *2.35 (3.94)* | *NS* |
| 14 | 1,984 | α,β-dihydroferulic acid, hydroxypropyl ester | 2.62 (2.47)[b] | 0.0121 (0.0270)[a] | 0.0297 (0.0425)[a] | <0.05 |
| | | | *1.64 (0.43)* | *0.47 (1.05)* | *1.11 (1.53)* | NS |
| 15 | 2,055 | 3-(3,5-dimethoxy-4-hydroxyphenyl)propionic acid propyl ester (hydrosinapinic acid propyl ester) | 1.14 (1.48)[b] | — | 0.0083 (0.0185)[a] | <0.05 |
| | | | *0.63 (0.37)[b]* | — | *0.23 (0.52)[a]* | <0.05 |

**Note:**
Data are mean chromatographic peak areas (standard deviation in parentheses), together with relative peak areas expressed as a mean % of the total across compounds (shown in italics). All identifications were confirmed with reference to standard compounds. LRI, linear retention index on a 15 m × 0.25 mm (0.25 μm film thickness) DB-5 MS capillary column. P = probability of a significant difference; Peak areas followed by the same letter are not significantly different (P > 0.05), while peak numbers refer to Fig. S1.

compound, β-(4-hydroxy-3-methoxyphenyl)propionic acid (α,β-dihydroferulic acid), was found in all groups and while higher in adult males, only reached a significant difference in adult males vs adult females. Table S1 provides supporting mass spectral data of the 15 compounds identified in the MCT extracts of fur.

We undertook headspace concentration of the solvent extract, to discover if there were compounds of interest present at low concentrations in the fur extract. Table 2 lists the

**Table 2 Headspace compounds in medium-chain triglyceride extracts of fur surrounding the shoulder gland of *Sturnira parvidens*, obtained using solid-phase microextraction.**

| Peak number | Linear retention index (LRI)[a] | Compound | Confirmation of identity[b] | Gas chromatographic peak areas (×10⁻⁶) | | | P |
|---|---|---|---|---|---|---|---|
| | | | | Adult male | Juvenile male | Adult female | |
| 1 | 904 | ethyl acetate | Std | 93.9 (17.9)[a] | 120 (10.8)[b] | 126 (9.31)[b] | 0.006 |
| | | | | *6.46 (2.27)[a]* | *17.86 (2.83)[b]* | *17.89 (1.85)[b]* | *<0.01* |
| 2 | 912 | isopropyl acetate | Std | 41.4 (9.89)[a] | 51.2 (7.70)[a,b] | 55.3 (4.87)[b] | 0.041 |
| | | | | *2.85 (1.08)[a]* | *7.64 (1.39)[b]* | *7.85 (0.60)[b]* | *<0.05* |
| 3 | 916 | 2-butanone | Std | 52.6 (18.6) | 61.6 (10.7) | 67.9 (9.64) | NS |
| | | | | *3.67 (1.81)[a]* | *9.19 (1.92)[b]* | *9.60 (0.87)[b]* | *<0.05* |
| 4 | 939 | isopropyl alcohol | Std | 76.4 (23.0) | 90.7 (8.95) | 96.8 (6.14) | NS |
| | | | | *5.52 (3.55)[a]* | *13.72 (3.35)[b]* | *13.93 (2.50)[a]* | *<0.05* |
| 5 | 947 | ethanol | Std | 115 (47.8) | 162 (43.5) | 154 (22.6) | NS |
| | | | | *7.40 (1.95)[a]* | *24.56 (8.64)[b]* | *22.14 (5.13)[b]* | *<0.05* |
| 6 | 972 | ethyl propanoate | Std | 9.78 (2.36) | 9.91 (1.95) | 10.7 (2.47) | NS |
| | | | | *0.68 (0.28)[a]* | *1.47 (0.30)[b]* | *1.57 (0.54)[b]* | *<0.05* |
| 7 | 1,030 | alpha-pinene | Std | 8.81 (0.994) | 12.6 (8.36) | 12.8 (12.2) | NS |
| | | | | *0.60 (0.17)* | *1.96 (1.57)* | *1.65 (1.31)* | *NS* |
| 8 | 1,038 | 2-butanol | Std | 17.0 (4.15)[b] | 9.02 (1.15)[a] | 7.06 (3.50)[a] | 0.001 |
| | | | | *1.13 (0.26)* | *1.34 (0.22)* | *0.98 (0.49)* | *NS* |
| 9 | 1,052 | 1-propanol | Std | 674 (114)[b] | 44.4 (73.7)[a] | 50.0 (65.4)[a] | <0.0001 |
| | | | | *44.58 (8.12)[b]* | *5.90 (9.32)[a]* | *6.23 (7.09)[a]* | *<0.01* |
| 10 | 1,096 | *N,N*-dimethylhydroxylamine | MS | 10.3 (9.53) | 4.40 (5.63) | 11.8 (10.9) | NS |
| | | | | *4.79 (9.17)* | *0.52 (0.65)* | *1.43 (1.15)* | *NS* |
| 11 | 1,132 | 2-pentanol | Std | 2.28 (1.35)[b] | 0.776 (0.078)[a] | 0.956 (0.156)[a,b] | 0.021 |
| | | | | *0.14 (0.04)* | *0.12 (0.01)* | *0.14 (0.02)* | *NS* |
| 12 | 1,138 | 3-methylthiophene | Std | 2.66 (2.25) | 5.83 (10.9) | 1.32 (1.45) | NS |
| | | | | *0.15 (0.11)* | *0.73 (1.34)* | *0.98 (0.49)* | *NS* |
| 13 | 1,140 | 1-methoxy-2-propanol | Std | 7.64 (4.21) | 8.36 (5.27) | 10.6 (3.71) | NS |
| | | | | *0.47 (0.16)[a]* | *1.25 (0.85)[a,b]* | *1.49 (0.52)[b]* | *<0.05* |
| 14 | 1,156 | 1-butanol | Std | 9.55 (6.42) | 4.66 (0.587) | 4.32 (1.03) | NS |
| | | | | *0.58 (0.22)[a]* | *0.69 (0.10)[b]* | *0.61 (0.11)[a,b]* | *<0.05* |
| 15 | 1,212 | limonene | Std | 5.19 (2.00) | 3.85 (3.25) | 3.68 (1.44) | NS |
| | | | | *0.33 (0.14)* | *0.54 (0.37)* | *0.53 (0.22)* | *NS* |
| 16 | 1,262 | 3-methyl-3-buten-1-ol | Std | 3.95 (3.15) | 3.28 (3.79) | 11.7 (15.7) | NS |
| | | | | *0.25 (0.14)[a]* | *0.43 (0.47)[a,b]* | *1.46 (1.69)[b]* | *<0.05* |
| 17 | 1,313 | 3-methylpyridine | Std | 1.17 (0.619)[b] | 0.248 (0.156)[a] | 0.403 (0.135)[a] | 0.005 |
| | | | | *0.08 (0.04)[a]* | *0.03 (0.02)[b]* | *0.06 (0.02)[a,b]* | *<0.05* |
| 18 | 1,404 | methyl octanoate | Std | 5.51 (5.69) | 6.18 (10.4) | 5.53 (5.17) | NS |
| | | | | *0.29 (0.24)* | *0.79 (1.26)* | *0.79 (1.26)* | *NS* |
| 19 | 1,448 | ethyl octanoate | Std | 3.04 (2.70) | 3.58 (4.77) | 4.84 (4.56) | NS |
| | | | | *0.17 (0.11)* | *0.47 (0.57)* | *0.69 (0.65)* | *NS* |
| 20 | 1,457 | (*E*)-linalool oxide furanoid | Std | 0.974 (1.08) | 0.011 (0.024) | 0.042 (0.040) | 0.05 |
| | | | | *0.05 (0.05)[b]* | *<0.00 (0.00)[a]* | *0.01 (0.01)[a]* | *<0.01* |

(Continued)

| Peak number | Linear retention index (LRI)[a] | Compound | Confirmation of identity[b] | Gas chromatographic peak areas (×10⁻⁶) | | | P |
|---|---|---|---|---|---|---|---|
| | | | | Adult male | Juvenile male | Adult female | |
| 21 | 1,474 | acetic acid | Std | 38.9 (14.7)[b] | 15.1 (7.80)[a] | 17.9 (3.65)[a] | 0.005 |
| | | | | 2.68 (1.41) | 2.15 (0.77) | 2.56 (0.57) | NS |
| 22 | 1,485 | (Z)-linalool oxide furanoid | Std | 3.98 (3.92)[b] | 0.043 (0.043)[a] | 0.080 (0.079)[a] | 0.026 |
| | | | | 0.24 (0.23)[b] | 0.01 (0.01)[a] | 0.01 (0.01)[a] | <0.01 |
| 23 | 1,533 | propyl octanoate | Std | 21.6 (31.2) | 0.360 (0.328) | 0.458 (0.400) | NS |
| | | | | 1.07 (1.45)[b] | 0.05 (0.04)[a] | 0.06 (0.06)[a] | <0.01 |
| 24 | 1,551 | benzaldehyde | Std | 2.32 (1.27) | 1.45 (0.167) | 1.54 (0.187) | NS |
| | | | | 0.16 (0.11)[a] | 0.21 (0.02)[b] | 0.22 (0.04)[b] | <0.01 |
| 25 | 1,558 | linalool | Std | 0.855 (0.753)[b] | 0.127 (0.029)[a,b] | 0.104 (0.062)[a] | 0.029 |
| | | | | 0.05 (0.04) | 0.02 (0.01) | 0.02 (0.01) | NS |
| 26 | 1,560 | propanoic acid | Std | 2.83 (0.843)[b] | 1.61 (0.753)[a] | 1.852 (0.318)[a,b] | 0.034 |
| | | | | 0.19 (0.06) | 0.23 (0.07) | 0.27 (0.06) | NS |
| 27 | 1,620 | propylene glycol | Std | 41.4 (39.3)[b] | 2.46 (0.598)[a] | 4.38 (1.24)[a,b] | 0.032 |
| | | | | 2.68 (2.74)[b] | 0.36 (0.06)[a] | 0.63 (0.21)[a] | <0.01 |
| 28 | 1,648 | butyric acid | Std | 3.62 (2.13) | 1.51 (0.967) | 2.31 (1.32) | NS |
| | | | | 0.24 (0.15) | 0.21 (0.11) | 0.33 (0.19) | NS |
| 29 | 1,663 | acrylic acid | Std | 9.29 (3.74) | 2.78 (3.58) | 7.40 (4.88) | NS |
| | | | | 0.59 (0.12)[a,b] | 0.37 (0.42)[a] | 1.01 (0.57)[b] | <0.01 |
| 30 | 1,757 | pentanoic acid | Std | 1.44 (1.27) | 0.527 (0.293) | 0.722 (0.263) | NS |
| | | | | 0.10 (0.11) | 0.07 (0.03) | 0.10 (0.04) | NS |
| 31 | 1,777 | (E)- or (Z)-linalool oxide (pyranoid) | Std | 0.926 (0.808)[b] | 0.006 (0.013)[a] | 0.025 (0.035)[a] | 0.013 |
| | | | | 0.05 (0.04)[b] | <0.00 (0.00)[a] | <0.00 (0.00)[a] | <0.01 |
| 32 | 1,847 | 3,4-dimethylbenzaldehyde | Std | 1.85 (0.362) | 2.16 (0.506) | 2.01 (0.421) | NS |
| | | | | 0.13 (0.04)[a] | 0.32 (0.06)[b] | 0.29 (0.07)[b] | <0.01 |
| 33 | 1,864 | hexanoic acid | Std | 4.04 (1.43)[b] | 1.68 (1.13)[a] | 1.80 (0.880)[a] | 0.012 |
| | | | | 0.26 (0.07) | 0.23 (0.12) | 0.26 (0.13) | NS |
| 34 | 1,886 | guaiacol | Std | 8.19 (7.91)[b] | 0.075 (0.029)[a] | 0.116 (0.063)[a] | 0.023 |
| | | | | 0.50 (0.54)[b] | 0.01 (0.00)[a] | 0.02 (0.01)[a] | <0.01 |
| 35 | 1,899 | benzyl alcohol | Std | 5.85 (8.05) | 0.210 (0.045) | 0.704 (0.502) | NS |
| | | | | 0.44 (0.67)[b] | 0.03 (0.00)[a] | 0.10 (0.07)[a,b] | <0.05 |
| 36 | 1,925 | propyl salicylate | Std | 1.02 (0.707)[b] | 0.024 (0.018)[a] | 0.027 (0.015)[a] | 0.003 |
| | | | | 0.07 (0.06)[b] | <0.00 (0.00)[a] | <0.00 (0.00)[a] | <0.01 |
| 37 | 2,031 | phenol | Std | 1.48 (0.382)[b] | 0.326 (0.089)[a] | 0.772 (0.252)[a] | 0.0001 |
| | | | | 0.01 (0.03)[a,b] | 0.05 (0.01)[a] | 0.11 (0.04)[b] | <0.01 |
| 38 | 2,056 | 5-ethyl-2-methoxyphenol | MS + LRI | 0.742 (0.563)[b] | 0.032 (0.023)[a] | 0.025 (0.006)[a] | 0.006 |
| | | | | 0.04 (0.02)[b] | <0.00 (0.00)[a] | <0.00 (0.00)[a] | <0.01 |
| 39 | 2,061 | 4-ethyl-2-methoxyphenol | Std | 0.551 (0.639) | 0.019 (0.011) | 0.023 (0.009) | NS |
| | | | | 0.03 (0.03)[b] | <0.00 (0.00)[a] | <0.00 (0.00)[a] | <0.01 |
| 40 | 2,076 | octanoic acid* | Std | 200 (302) | 49.8 (81.0) | 32.9 (28.7) | NS |
| | | | | 9.78 (14.14) | 6.43 (9.76) | 4.71 (4.17) | NS |

| Peak number | Linear retention index (LRI)[a] | Compound | Confirmation of identity[b] | Gas chromatographic peak areas ($\times 10^{-6}$) | | | P |
|---|---|---|---|---|---|---|---|
| | | | | Adult male | Juvenile male | Adult female | |
| 41 | 2,197 | 4-vinyl-2-methoxyphenol | Std | 1.01 (0.635)[b] | 0.032 (0.025)[a] | 0.038 (0.022)[a] | 0.002 |
| | | | | *0.06 (0.03)[b]* | *<0.00 (0.00)[a]* | *0.01 (0.00)[a]* | *<0.01* |
| 42 | >2,200 | anthranilic acid propyl ester | Std | 5.71 (4.55)[b] | 0.150 (0.054)[a] | 0.222 (0.124)[a] | 0.008 |
| | | | | *0.35 (0.30)[b]* | *0.02 (0.01)[a]* | *0.03 (0.02)[a]* | *<0.01* |

Notes:

Data are mean chromatographic peak areas (standard deviation in parentheses), together with relative peak areas expressed as a mean % of the total across compounds (shown in italics). P = probability of a significant difference; Peak areas followed by the same letter are not significantly different (P > 0.05), while peak numbers refer to Fig. S2.

[a] Linear retention index on a 30 m × 0.25 mm (0.5 μm film thickness) Stabilwax DA capillary column.

[b] Std, standard compound run under the same conditions; MS + LRI, mass spectrum and linear retention index similar to literature spectrum (*Schranz et al., 2017*); MS, mass spectrum similar to literature spectrum (NIST11.L for Chemstation).

* Octanoic acid is also contributed by breakdown of the MCT solvent.

headspace compounds in MCT extracts of fur surrounding the shoulder gland of *Sturnira parvidens*, obtained using SPME, and representative chromatograms are shown in Fig. S2. These are the more volatile components in the extracted glandular secretion from *Sturnira parvidens* shoulder gland fur. We identified 42 compounds, 33 showing significant differences between adult males, juvenile males and adult females for peak area (six compounds), percentage peak area (14 compounds) or both (13 compounds). A total of 21 peaks were higher and 11 peaks were lower in adult males, compared to females and juveniles. One peak was higher in females vs juvenile males for percentage peak area (acrylic acid). Collectively these results indicate that adult male shoulder gland secretion odour profiles are distinct from those of females and juvenile males. The 12 peaks with the largest area that were significantly greater in males compared to females and juvenile males were 1-propanol, acetic acid, 2-butanol, guaiacol, anthranilic acid propyl ester, (*Z*)-linalool oxide furanoid, phenol, 3-methylpyridine, propyl salicylate, 4-vinyl-2-methoxyphenol, (*E*) *or* (*Z*)-linalool oxide (pyranoid), and 4-ethyl-2-methoxyphenol, respectively. Linalool was approximately eight times higher in males (mean peak area) than females and juvenile males but was only significantly different vs females.

Figure 2 shows the overall odour fingerprint analysis as NMDS plots for direct injection and headspace extracts, and for both the absolute and relative percentage peak areas. In all cases, adult male profiles were distinct from those of adult females and juvenile males, with the latter showing an overlapping pattern. Multivariate statistical analysis of similarity of the relationship between grouping variable (adult and juvenile males and adult females) and chemical composition of fur extract showed a significant effect of group (Headspace ANOSIM on Bray–Curtis similarity matrix, $R = 0.4922$, $P \leq 0.001$, Direct Injection ANOSIM on Bray–Curtis similarity matrix, $R = 0.5911$, $P \leq 0.001$).

## DISCUSSION

Odour fingerprint analysis of extracts of fur surrounding the shoulder gland of *Sturnira parvidens*, using multivariate analysis, revealed clear, significant differences between adult males vs both juvenile males and adult females. Many of the compounds identified included terpenes and phenolics, together with alcohols and esters, and occurred

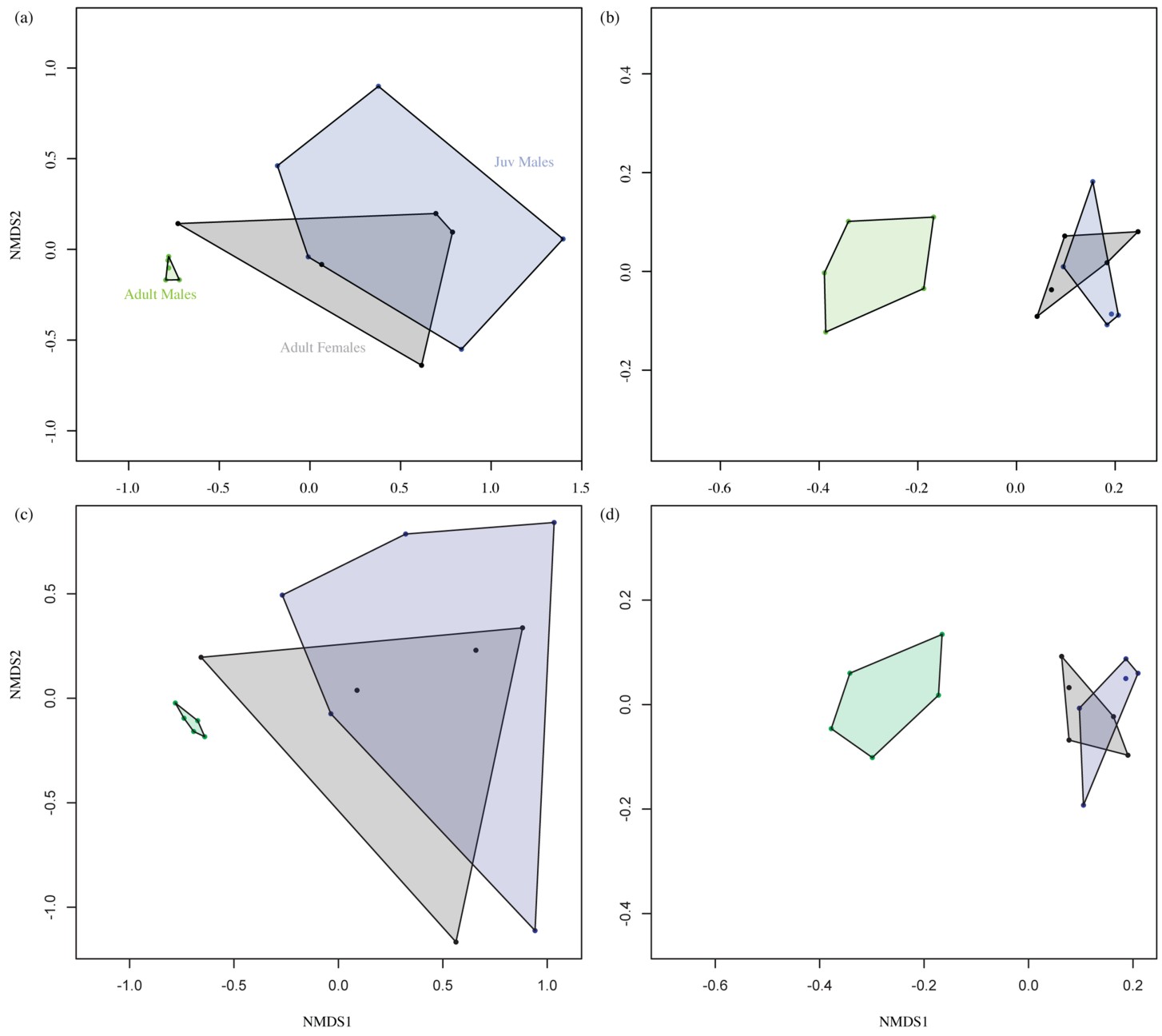

**Figure 2 Non-metric multidimensional scaling (NMDS) plot.** Data represent the overall odour profiles of (A) and (C) direct injection, (B) and (D) headspace extracts from the shoulder gland secretions of adult males, adult females and juvenile (Juv) males. Profiles (A) and (B) are generated from the relative percentage peak area, while (C) and (D) are from absolute peak areas. Adult male profiles are distinct from those of females and juveniles, which show an overlapping pattern. Axes are dimensionless and have no units.

exclusively or in elevated quantities among adult (reproductive) males, compared with adult females and non-reproductive sub-adult/juvenile males. This strongly suggests a specific role in male–female attraction for these compounds, although a function in male–male competition and/or species recognition is also possible. While non-reproductive responses have been attributed to pheromones (e.g. aggression and alarm signals), sexual stimulation and attraction are the most commonly associated traits

of pheromones (though the two responses can happen simultaneously to the same substance for different individuals (*Novotny, 2003*)).

Among animals where vision is not a primary sensory modality, chemical cues have been associated with species recognition. Many species of bat are highly gregarious, forming large, often multi-species colonies that are stable units (*Kunz, 1982*), and chemical recognition is known to assist in mother–pup recognition (*De Fanis & Jones, 1996*) and colony identification (*Bloss et al., 2002*). Although relatively little is known about *Sturnira* social systems, they are not known to roost in the large groups or colonies typical of many bats, but rather in small groups in tree holes or other small cavities (*Fenton et al., 2000*). However, the distinctiveness of the shoulder gland odour profiles that we have characterised for reproductively active adult males suggests strongly that they are more likely to be involved in mating or mate choice. A small number of other studies among bats have suggested a role for odour communication in individual and social group recognition. Female big brown bats (*E. fuscus*) were able to use chemical cues to distinguish roost mates and in a Y-maze experiment chose to preferentially associate with the familiar cue (*Bloss et al., 2002*). In female Bechstein's bats, *Myotis bechsteinii*, odour profiles from secretions of the facial interaural gland are individually specific and differ between colonies, suggesting a function in individual and colony recognition (*Safi & Kirth, 2003*). A study of the distinctive odour of the fisherman bat (*N. leporinus*), produced by a secretion from glands in the sub-axillary region beneath the wings, identified 372 lipid compounds (*Brooke & Decker, 1996*). Differences in the chemical composition suggested that secretions of males from the same roost were more similar to each other than to other males or females. Furthermore, as secretions differed between the sexes, information on sexual identity and reproductive condition could potentially be communicated. As with *Sturnira parvidens*, another phyllostomid, the male greater spear-nosed bat, *Phyllostomus hastatus*, also possesses a sexually dimorphic gland that produces an odoriferous secretion, but it is found on the chest rather than the shoulder. A recent study by *Adams, Li & Wilkinson (2018)* reports that secretions from male harem holders, that defend and roost with groups of females, had significantly different chemical profiles from bachelor males found roosting in all male groups. Odour profiles also differed significantly among individuals, suggesting that the chemical signal has the potential to communicate both mating status and individual identity.

While the aforementioned provide good evidence for odour signals/pheromones playing an important role in reproductive and social behaviour in bats, our study goes further in identifying the chemical composition of the *Sturnira parvidens* odour signal. In the adult male shoulder gland secretion, the components include terpenes and phenolics, together with alcohols and esters. It is likely that these are derived from the frugivorous diet of *Sturnira parvidens*. *Bloss et al. (2002)* identified 15 potential chemicals associated with female *E. fuscus* and, of these, linalool was also found in our analysis of *Sturnira parvidens*, together with three linalool derivatives, all of which were elevated in reproductive males. Linalool is a terpene alcohol naturally produced by over 200 plant species, and there are many examples of it functioning to attract insects to plants (e.g. hawkmoths; *Raguso & Light, 1998*), and is the mate attractant pheromone in the bee
*Colletes cunicularius* (*Borg-Karlson et al., 2003*), suggesting a very broad biological action across plant and animal kingdoms.

Phenolics found in higher concentrations in the adult male secretions in our analysis include vanillic acid (a natural phenol found, for example, in palm fruit), and guaiacol. The latter is a component of the pheromone that causes locust swarming (*Dillon, Vennard & Charnley, 2000*) and produced in the gut of desert locusts, *Schistocerca gregaria*, by the breakdown of plant material. Phenol itself was also identified as a major component in our adult male samples, and interestingly is also part of the temporal gland secretion of adult male elephants, which is thought to function to attract females during muste, the periodic aggressive behavioural condition exhibited by bull elephants (*Rasmussen & Perrin, 1999*). We also identified esters of ferulic acid, the latter being a phenolic antioxidant found in the seeds of apples and oranges and the cell walls of many plants.

In addition to vanillic acid, we identified a number of other phenolic acids and propyl esters derived from them (Table S2). These included dihydroferulic, hydroisoferulic, hydrosinapinic, veratric and anthranilic propyl esters and the hydroxypropyl esters of dihydroferulic acid and anthranilic acid (Table 1). Interestingly, α,β-dihydroferulic acid propyl ester has been shown to exhibit strong antifungal action, perhaps suggesting a role beyond just odour signalling for this compound (*Beck et al., 2007*). Significant quantities of 1-propanol were found in the headspace analysis of the adult male compared to other alcohols. By contrast, only small quantities of 1-propanol were found in the headspace of the juvenile male and the female, suggesting that it and/or the propyl esters may play a role in pheromonal signalling in this bat species. 1-propanol may either be acting as a chemical signal in its own right, or produced to form the aforementioned esters with the various phenolic acids. All the acids and esters found in the adult male extract were at significantly higher levels than in the juvenile male and the female. Additionally, anthranilic acid propyl ester was also identified in the headspace analysis (Table 2) at significantly higher levels in the adult male than in the juvenile male and the female. The hydroxypropyl esters of dihydroferulic acid and anthranilic are derived from propylene glycol, which was also identified in significantly higher quantities in the headspace analysis of the adult male, compared to the juvenile male and the female, suggesting that these esters may also have pheromonal or semiochemical activity. Three of these esters (Table 2) have not been previously reported and are unique to *Sturnira parvidens*.

Bacteria have been associated with the production of many chemical cues and it is possible that these may also play a role in the synthesis of some of the compounds that we identified. In turn, these may contribute to individual and population-specific odour signatures in *Sturnira parvidens* (if populations share local communities of bacteria in their shoulder glands). In hyenas, bacterial species co-vary with odour profiles specific to populations, supporting such a 'fermentation hypothesis' of bacterial mediated chemical communication (*Theis et al., 2013*). An investigation of bacteria associated with the epaulettes (shoulder glands) of the related bat species *Sturnira lilium* and *S. bogotensis* revealed few common bacterial species between males and females, offering the possibility for sex-specific odour production in the glands of males (*González-Quiñonez, Fermin & Muñoz-Romo, 2014*).

## CONCLUSIONS

In summary, many of the compounds we identified from our analysis were found exclusively or in elevated quantities among adult (reproductive) males, compared with adult females and non-reproductive males. Similar analyses of other *Sturnira parvidens* populations and other *Sturnira* species would be interesting, enabling a comparison of the variance in the composition of the odour signal. Our results strongly suggests a specific role in male–female attraction in this species, but further behavioural work is needed to confirm the functional significance of the adult male shoulder gland secretions in *Sturnira parvidens*.

## ACKNOWLEDGEMENTS

We thank the staff of the Lamanai Outpost Lodge, especially Mark Howells, Blanca Manzanilla, Ruben Arvalo, and Eduardo Ruano for logistical support. We further thank the many field teams on the Belize bat trip for contributing to data collection over many years.

### Funding

Funding for fieldwork was provided by the Taxonomic Mammalogy Fund of the American Museum of Natural History. The funders had no role in study design, data collection and analysis, decision to publish, or preparation of the manuscript.

### Grant Disclosures

The following grant information was disclosed by the authors:
Taxonomic Mammalogy Fund of the American Museum of Natural History.

### Competing Interests

David A. Baines is employed by Baines Food Consultancy Ltd.

### Author Contributions

- Chris G. Faulkes conceived and designed the experiments, analysed the data, prepared figures and/or tables, authored or reviewed drafts of the paper, approved the final draft.
- J. Stephen Elmore conceived and designed the experiments, performed the experiments, analysed the data, contributed reagents/materials/analysis tools, prepared figures and/or tables, authored or reviewed drafts of the paper, approved the final draft.
- David A. Baines conceived and designed the experiments, contributed reagents/materials/analysis tools, authored or reviewed drafts of the paper, approved the final draft.
- Brock Fenton performed the experiments, authored or reviewed drafts of the paper, approved the final draft.
- Nancy B. Simmons performed the experiments, authored or reviewed drafts of the paper, approved the final draft.

- Elizabeth L. Clare conceived and designed the experiments, performed the experiments, analysed the data, contributed reagents/materials/analysis tools, authored or reviewed drafts of the paper, approved the final draft.

## Animal Ethics

The following information was supplied relating to ethical approvals (i.e. approving body and any reference numbers):

Our research was conducted in accordance with accepted standards for humane capture and handling of bats published by the American Society of Mammalogists and approved Institutional Animal Care and Use Committee protocols (Brown University IACUC 1205016 and 1504000134).

## Field Study Permissions

The following information was supplied relating to field study approvals (i.e. approving body and any reference numbers):

Field experiments were approved by the Ministry of agriculture, fisheries, forestry, the environment and sustainable development, Belize Forestry Department (Scientific Research and Collecting Permits CD/60/3/15 (20) and WL/1/1/16 (26))

## Data Availability

The raw data are available in Tables 1, 2, and Table S1.

## Supplemental Information

Supplemental information for this article can be found online at http://dx.doi.org/10.7717/peerj.7734#supplemental-information.

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
