# Peer review of "Chemical characterisation of potential pheromones from the shoulder gland of the Northern yellow-shouldered-bat, Sturnira parvidens (Phyllostomidae: Stenodermatinae)"

_PeerJ, doi:10.7717/peerj.7734_

## Round 0.1 · original submission · Minor Revisions

We have received two expert reviews who feel the manuscript requires minor revisions. I agree that there are several areas requiring clarification and additional information (mainly for the purposes of replicating the results in other studies). While each reviewer has provided a detailed accounting of flagged issues, much of this is text-based. Please pay careful attention, in particular, to the questions and requests that will clarify understanding of the paper along with the more routine requests for specific edits. Be sure to address each issue in your response.

Reviewer 1 ·

Basic reporting

Overall, this manuscript contains interesting data that certainly improve our knowledge of chemical signals in bats. Although the language and tone are appropriate, the presentation lacks some detail and clarity.

Pheromones have been defined in a variety of ways and the authors clearly state their working definition, but given the semantic debate over this term, it may be best to avoid it altogether. Since the function of this chemical signal is still unknown, it may be better to use a more general term like semiochemical or chemical signal. Additionally, the description of the two categories of pheromonal effects (lines 49-55) is curious as it seems to conflate different classification schemes. Further, the definition of primer pheromones is inconsistent with previous definitions (see Wyatt 2010). Rather than try to classify semiochemicals, it might be more useful to give a general sense of diversity of functions. This issue of defining pheromones reappears in the discussion (e.g. lines 287-290). It may be true that sexual stimulation and attraction are the most common functions assigned to pheromones, but perhaps this is because the term pheromone is more often invoked for chemical signals involved in reproduction. There numerous examples of mammalian chemical signals that are not directly used for mate attraction and stimulation.

The introduction incorporates some relevant background, but the overall organization could be improved. For example, in the second paragraph the authors make a case for the importance of chemical signaling in bats (although citations are sparse), but the details of variation in vomeronasal organ and associated genes seems out of place. As the authors mention later, the absence of the VNO does not preclude the use of intra-specific chemical signals. The variation among bats is interesting, just not relevant at this point in the manuscript. If the authors insist on keeping this content in the introduction, it would be helpful first introduce the structural variation, then discussion the genetic variation. Wible and Bhatnagar (Journal of Mammalian Evolution, 1996, 3:285-314) described variation in VNO morphology before the more recent genetic methods.

I appreciate at the introduction concludes with a clear statement of their objective and brief statement of the methods used. This objective is clearly met with the data presented. With a bit of polishing this manuscript will be a great foundation for further study of chemical communication in Sturnira spp.


Detailed remarks:
line 27: Consider rewording “mostly highly developed” to “more developed”
line 81: “A number of species in the family Emballonuridae [have] sacs…”
line 86: This description of S. bilineata behavior seems to conflate salting with hover displays.
line 105: Consider replacing “unstudied” with “understudied” as the latter half of the paragraph shows the trait is not completely unstudied.
line 95: Much of the work on S. bilineata described here was done prior to Schneeberger et al 2016. Also, the chemical composition has been examined by Caspers et al (Chemical Signals in Vertebrates 11, 2008, pp. 151-160).
line 126: extra parenthesis in citation
line 150: ambiguous sentence
line 175: “We carried out [v]olatile…”
line 228: It should be Bray-Curtis dissimilarity
line 244 (and elsewhere): The phase ‘status groups’ is initially confusing. Define clearly.
line 315: Intended meaning of “specificity” is unclear. Consider rewording to emphasize sexual dimorphism.
line 334: sentence fragment
line 349: cumbersome and potential run-on. Consider splitting.
line 392: “…further work [is] needed to confirm…”
line 408: Bhatnagar and Meisami (1998) appears in the reference list but is not cited in the text.

Experimental design

The overall design of the study seems sound and the methods used to identify the individual compounds are rigorous, but several details are missing from the description of the methods. For example, during what months were the samples collected? If these bats are seasonal breeders, one might expect seasonal variation in such a sexually dimorphic secretion. Given that males are classified as either reproductive adults or sub-adults, it would be helpful to have more detail about the assessment of reproductive status. Are all adults assumed to be reproductive or was the size/state of the testes examined?

The description of where the samples were collected from the bats is unclear (line 149-151). At first I thought two samples were collected, one from the shoulder patch and one from the surrounding area, but later realized only a single sample type was analyzed. Line 257 refers to the shoulder gland fur, whereas most other instances refer to the fur surrounding the gland. This may be semantic nit-picking, but the location of the hair samples is critical to the study.

Given that some solvents can interact with plastics, what type of vials were used to store the samples? Where any blanks (same solvent and vial but without the hair sample) run as controls? Also, the authors note that the MCT solvent stabilizes the secretion, but it should still be reported approximately how long the samples were stored before analysis. On a similar note, how well does the solvent stabilize the secretion, if volatiles were able to be extracted from the headspace via SPME?

The chosen statistical methods are reasonable, but the large differences in dispersion between groups may impact the results (see Anderson and Walsh. Ecological Monographs, 83(4), 2013, pp. 557–574).

Some additional details will also allow the reader to better understand how the data were analyzed. For example, the authors report there were 15 major peaks, but is unclear what criteria were used to define peaks as major. Perhaps this distinction is obvious from the chromatograms, but a sample chromatogram was not provided. Similarly, it seems the relative areas are calculated relative to only the total area of the major peaks not the total area, both major and non-major; this should be made more clear. The software that was used to identify peaks and calculate the area should also be indicated.

It is also unclear to me why the authors have chosen to analyze both relative and absolute areas. It’s completely understandable that the size of the hair sample could be precisely controlled, and using relative areas is an obvious solution, so why also analyze absolute areas at all? From Table 1 it appears that males consistently had higher peak areas and this may be interesting in its own right. If the fur samples were approximately similar in size, but the adult males consistently had much stronger signals, that may suggest adult males have more active glands. This is qualitatively interesting, even if it can be rigorously quantified. When examining composition, it seems more appropriate to use relative abundances. It is unclear which set of responses (absolute or relative) was used for the ANOSIM. Additionally, the number of permutations used in both the ANOSIM and SIMPER procedures should be reported.

The authors clearly state their compliance with ethical guidelines and animal handling protocols.

Validity of the findings

The authors do a nice job of connecting their findings to other studies that found similar compounds in semiochemicals of other species as well as dietary components. This was a clear strength of the discussion. Overall the discussion of the findings is appropriate, with one exception. The authors claim that their findings “strongly suggest a specific role in male-female attraction” (lines 40 & 285). This is not necessarily true. The sexual dimorphism of the glandular product does suggest a role in mating, but it is just as likely used for male-male competition as it is for mate attraction. At this point, one also cannot rule out its role in species recognition.

On lines 36 and 281 the authors suggest that the NMDS was the analysis used to test for differences. This is incorrect. NMDS is helpful for visualization, but the statistical significance is derived from the multivariate analysis of similarity.

Additional comments:
line 238: The text implies that only 5 samples were collected, when it seems that 5 samples were collected from each age/sex class.
line 237- 238: This sentence implies that all 15 peaks differed significantly, but that is not true.
line 259: sample size numbers are confusing because they refer to numbers of peaks, not samples

·

Basic reporting

see general comments

Experimental design

no comment

Validity of the findings

no comment

Additional comments

This is an interesting paper which reports a first chemical characterisation of secretions which may have semiochemical roles in the bat Sturnia parvidens. As the authors observe, the differences between the secretions of adult reproductive males, juvenile males, and females indicate that these differences may be important in courtship and even mate choice in adults. The potential roles of these semiochemicals will no doubt be the focus of future investigations.

However, the introduction perhaps over emphasises novelty (e.g. Line 57 “yet very little is known about the possible role of pheromonal communication in this group” and Line 63 “ little work on olfactory systems in bats “) even though novelty is not a criterion for publication in peerJ. The current ms is an interesting piece of work in its own right. The introduction could be constructively rewritten to help the reader place the paper in the context of the fairly extensive previous work on semiochemicals in bats, currently cited later in the Introduction and in the Discussion, bringing them early into the Introduction. None of the previous work detracts from the interest of the current paper which builds on the solid work of others.

More specific points follow.

1. Line 49 ff Rewording vs signature mixtures/pheromones with priming effects to remove the confusion of the terms. The concepts are perhaps more clearly explained in my 2017 short review [Wyatt, TD (2017) Pheromones. Current Biology 27: R739–R743.] “Many pheromones … have both immediate ‘releaser’ effects and longer lasting primer effects.“ and “It is worth separating the concepts of pheromones and signature mixtures (it was a mistake in my 2003 book to combine them). It is precisely because individuals have different odour profiles that these can be learnt and used to distinguish different animals.”

The current text reads: Line 49 ff “There are two general categories of pheromonal effect: primer pheromones are fixed chemical signals that elicit a defined response in the recipient animal (behavioural or physiological), while signature mixtures are emitted signals which are learned by the conspecific receivers and often have complex and variable chemical profiles. These “individual mixtures” or “signature odours”, which act in social communication though learning, rather than eliciting a specific behavioural or endocrine response, may identify a specific individual or social group (Wyatt 2010, Dehnhard 2011).”

Suggest change to “ “There are two general categories of semiochemicals within a species: pheromones, chemical signals which elicit a stereotypical behavioural or physiological response in the recipient animal, while signature mixtures are odour cues which are learned by the conspecific receivers and often have complex and variable chemical profiles. These “individual mixtures” or “signature odours”, which act in social communication though learning may identify a specific individual or social group (Wyatt 2017, 2014, Dehnhard 2011).”

2. Pheromones (in mammals) are not exclusively detected by the VNO (see Section 9.3.1 in Wyatt 2014 or Figure 4 in Wyatt 2017) [as noted in the ms Lines 127-8]. It would be worth also adding here, after “potential pheromones.” , “Pheromones in mammals are also detected by the main olfactory system (Wyatt 2014, 2017) so the lack of functioning VNO receptors does not rule out the possibility of pheromones.”

3. Could the authors integrate the information about bat VNO in the paragraph starting Line 56 with the paragraph starting Line 116. (in the later paragraph we learn that Sturnia parvidens does have a VNO and functional Trpc2.)

4. Line 97 “did not undertake actual chemical characterisation of the wing-sac liquids” does not seem to match the cited paper [Schneeberger et al. (2016)] which reports in its methods:
“Wing sacs were wiped out with a piece of cotton of ∼5 mg that had previously been washed with dichloromethane (99.9%) to remove potential contaminating substances. We stored the samples in a Teflon-capped glass vial (2 ml RotilaboH, Karlsruhe, Germany) at -80◦C until analysis. After sample collection, we released all bats at the site of capture.
[…]
Chemical Analysis
All samples were prepared for analysis and analyzed at Bielefeld University. Prior to chemical bouquet extraction, we added 100 µl of dichloromethane (DCM) to each sample”

5. Line 285: “This strongly suggests a specific role in male–female attraction (rather than just species recognition” . A sex pheromone can have both functions.

6. Line 287 ff As there has been no behavioural or physiological investigations of the secretions yet, I would delete Lines 287-306 inclusively. Such speculation is not helpful to the reader.

7. Line 308 Suggest delete the sentence starting “For example, in mixed”.

8. Line 329 – description of P hastatus and its sexually dimorphic glands might be better in the introduction? Interesting.
Discussion

9. Lines 337-376. Delete the words comparing the molecules found in this species with those of unrelated species. Whether a molecule is a pheromone in another species is largely irrelevant (unless it’s a close relative). Tells us nothing about the likely importance in this species.

10. Line 393 – suggest change “odour signatures” to “secretions”

11. Minor: Wyatt 2010 missing from reference list

---

## Round 0.2 · Minor Revisions

The revised article is much improved and almost ready to be recommended for publication. One reviewer raises a few points that should be addressed including an additional figure for headspace analysis (including tips on how to simplify the data for the figure), and clarification of ANOSIM results, and a few typos. Please make these very minor changes and the article will be ready for acceptance.

Reviewer 1 ·

Basic reporting

The revised manuscript is greatly improved and I look forward to seeing it published. The revised introduction provides adequate background and the discussion addresses how the results fit within the current state of knowledge.

Experimental design

The methods now provide sufficient detail to fully understand the procedure and replicate the results. Supplemental figure 1 is a nice addition, as it really highlights how distinct the adult males' profiles are, relative to females and juvenile males. It would be helpful to see a similar figure for the headspace analysis as well. Labeling all the peaks may not be feasible, given the number of compounds listed in table 2, but perhaps those that differ significantly can be indicated.

It is still unclear whether the reported ANOSIM results are based on absolute or relative peak areas. Figure 2 contains four plots, but only two statistical results are included in the text. The methods seem to imply the relative abundances were used, but this should be stated more clearly.

In the methods, the authors state they use a SIMPER analysis to identify which compounds underlie the differences between the groups; however, the footnote of Table 2 states the significance was assessed by a one-way ANOVA. Are the differences reported in the text based on the SIMPER analysis as suggested by the methods or one-way ANOVAs as suggested by the table footnote?

Validity of the findings

The revisions adequately address the initial over-emphasis on mate attraction.

Additional comments

Line 195: missing parenthesis
Line 307: replace "using non-metric multi-dimensional scaling" with "multivariate analyses"

·

Basic reporting

no comment

Experimental design

no comment

Validity of the findings

no comment

Additional comments

The authors have made significant and helpful improvements to the ms in response to the reviewers' comments. I particularly appreciated the detailed responses and the availability of the tracked-changes ms version which made it quick to assess revisions.

---

## Round 0.3 · accepted · Accept

Thank you for addressing the remaining concerns. The manuscript is now ready for publication.